# ONLINE TEST-TIME ADAPTATION IN TABULAR DATA WITH MINIMAL HIGH-CERTAINTY SAMPLES

## ABSTRACT

Tabular data is ubiquitous across real-world applications. While self-supervised learning has advanced representation learning for tabular data, most methods assume the unrealistic IID setting. In practice, tabular data often exhibits distribution shifts, including both label and covariate shifts, rendering existing domain generalization or test-time adaptation techniques from computer vision ineffective. To address this, we propose a simple yet effective **O**nline **T**est-**T**ime **A**daptation approach for **T**abular data (OT3A). It leverages high-confidence and domain-consistent pseudo-labels to estimate and correct for target label distribution shifts. Subsequently, it employs self-training and entropy minimization, guided by these confident samples, to adapt the model to the out-of-distribution test data. Extensive experiments across diverse distribution shift scenarios demonstrate that OT3A significantly outperforms existing methods, highlighting its efficacy and practicality for adapting to real-world tabular data. *Our code will be released in the supplemental material and will be open to the public in the future.*

## 1 INTRODUCTION

Tabular data is widely utilized across numerous real-world application scenarios (Zhou et al., 2018; Guo et al., 2021; Chen et al., 2016; Sadar et al., 2023; Abdou & Pointon, 2011). In recent years, deep neural network models (Chen et al., 2022; Yoon et al., 2020; Popov et al., 2020; Huang et al., 2020; Somepalli et al., 2021), have demonstrated remarkable performance in the field of self-supervised learning. However, these models are developed and trained under the assumption that the training and test data follow an independent and identically distributed (IID) paradigm. In the real world, structured tabular data often encounters the issue of distribution shift. Over time or due to environmental changes, the distribution of the test set of tabular data is likely to differ significantly from that of the training set. Such discrepancies can lead to a substantial degradation in the model's performance when deployed in real-world scenarios. To address these challenges, Test-Time Adaptation (TTA) has emerged as a promising solution for handling distribution shifts in structured tabular data.

Test-Time Adaptation (TTA) has been widely applied across various domains, such as computer vision (Sun et al., 2020b; Wang et al., 2021; Niu et al., 2022), natural language processing (Hardt & Sun, 2024; Banerjee et al., 2021), and speech processing (Bai et al., 2023; Huang et al., 2024). However, the application of TTA to tabular data remains largely unexplored. When existing TTA methods from other domains are directly applied to tabular data, their performance is often suboptimal. In this work, we conduct an in-depth analysis of this phenomenon and identify two primary underlying challenges: (1)- High imbalance in tabular data: Tabular data often exhibits highly imbalanced class distributions, which leads to model predictions during the test phase frequently being biased toward the source (training) domain distribution. This bias hinders the model's ability to adapt to the true distribution of the target domain, resulting in a significant performance degradation. (2)- Complexity of distribution shifts in tabular data: Distribution shifts in tabular data typically involve a mixture of multiple types of shifts, such as covariate shift, label shift, and concept drift. Existing methods, however, are usually based on a single underlying assumption (e.g., only considering label shift or covariate shift). Such approaches fail to effectively handle the complex and diverse nature of distribution changes in tabular data, thus limiting their effectiveness.

In real-world tabular data, the distribution at each time may dynamically change. To address this, we adopt OTTA (Online Test-Time Adaptation) (Azizzadenesheli et al., 2019; Park et al., 2023;

Alexandari et al., 2020), a stepwise adaptation framework. During the testing phase, OTTA processes incoming data in batches and makes online adjustments to the model based on each new test sample. Pseudo-label learning and entropy minimization are commonly used methods for OTTA. Pseudo-label based approachs (Lee, 2013) assigns pseudo-labels to unlabeled target domain data and uses these pseudo-labeled samples for adapting the pre-trained model to the target domain. Entropy minimization is a commonly used loss function in test-time adaptation (TTA). Related studies (Niu et al., 2022; 2023) have shown that low-entropy samples contribute more to model adaptation compared to high-entropy samples. Focusing on low-entropy samples can effectively accelerate the convergence of the model to the target domain distribution. We have defined high consistent and confidence sample points in tabular data. The pseudo-labels of these sample points have high certainty and can be effectively used to guide model adjustments.

In this work, we propose a novel uncertainty-based pseudo-labeling framework for **O**nline **T**est-**T**ime **T**abular **A**daptation, termed OT3A. Our method leverages high-confidence and domain-consistent pseudo-labels to estimate and correct for target label distribution shifts. Subsequently, it employs self-training and entropy minimization strategies, guided by these confident samples, to adapt the model to out-of-distribution test data. OT3A operates in an online algorithm: the model updates sequentially as each test mini-batch arrives, which is crucial for dynamic environments where data distribution may drift over time. Our main contributions are as follows:

- We investigate the OTTA problem for tabular data with detailed analysis to highlight its core challenges: The co-existence of label and covariate distribution shifts, and even class imbalance.

- We propose a novel and practical method, OT3A, which leverages high-certainty samples to estimate the label distribution and integrates entropy minimization with pseudo-label learning to simultaneously address both label and covariate shifts.

- We demonstrate through extensive experiments that OT3A achieves significant performance improvements over existing methods under various distribution shift settings.

## 2 PRELIMINARY

### 2.1 PROBLEM SETUP

In the online test adaptation problem of tabular classification, the input space is $X \in \mathbb{R}^d$, where $d$ represents the number of features, and each feature can be a continuous or discrete value. The label space is $Y \in \{0, 1\}^K$, where $K$ is the number of classes. We assume that we can access a pre-trained source model and adjust this model for test inputs during testing to make final predictions. The source model $f_{\theta_S}$ parameterized by $\theta_S$ is pre-trained on a labeled source domain $D_S = \{(x_S, y_S)\}$, and the source domain is formed by i.i.d. (independent and identically distributed) sampling from the source distribution $p_S$. The unlabeled test data $D_T = \{x_T^1, x_T^2, x_T^t, \ldots, x_T^n\}$ arrives in batches, where $t$ represents the current time step and $n$ is the total number of time steps (i.e., the number of batches). During the Test Time Adaptation (TTA) process, we update the model parameters for the $t$-th batch to obtain an adapted model $f_{\theta_t}$. At the $t$-th step, the received dataset is denoted by $D_T^t = \{x_i^t | 1 \leq i \leq n\}$, where the received instance $x_i^t \in X$ and its unobserved correct label $y_i^t$ are from the distribution $p_T(x, y)$ that is misaligned with $p_S(x, y)$, that is, $p_T(x, y) \neq p_S(x, y)$. The goal of Online Test Time Adaptation (OTTA) is that the model no longer depends on any known label information. Instead, by observing the unlabeled test data $D_t$ arriving in batches, the model iteratively updates its parameters from $\theta_t$ to $\theta_{t+1}$ at each time point $t$ to adapt to the new data distribution.

### 2.2 PROBLEM ANALYSIS

**The co-existence of covariate and label shifts of TTA in tabular data.** We analyzed distribution shifts across multiple benchmark datasets, employing the Wasserstein distance to measure covariate and label distribution shifts, respectively, as shown in Figure 1 (a). The results indicate varying degrees of shift across different datasets. Specifically, the HELOC and ASSISTments datasets exhibit significant shifts in both covariate and label distributions, suggesting substantial differences in their underlying data characteristics. In contrast, the Hospital Readmission and Voting

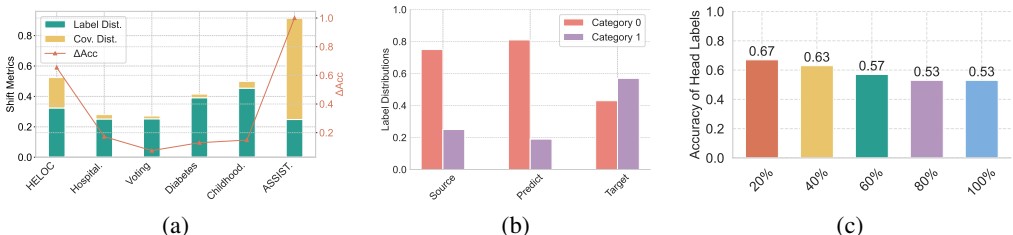

Figure 1: (a) The label and covariate distribution shifts between IID and OOD testing in tabular data. (b) Label distribution of source, target, and predicted (pseudo-labeled) samples on HELOC. (c) Accuracy of high-confidence predictions on HELOC, showing superior performance compared to all data.

datasets show smaller shifts, indicating greater consistency in their data distributions. In addition, The observed variations in shift magnitude likely represent a key factor contributing to performance challenges. For example, the transition from DIABETES to HELOC is accompanied by an increase in label shift, which correlates with decreased performance on test data. Similarly, the shift from HELOC to ASSIST, characterized by a pronounced increase in covariate shift, also confirms that distribution changes are detrimental to model generalization. While existing robust TTA methods (such as ODS (Zhou et al., 2023)) aim to mitigate the impact of such shifts, experimental results demonstrate their limited effectiveness. The diversity and varying degrees of shifts across different datasets likely explain why current TTA techniques struggle to effectively address both types of distribution change simultaneously in tabular data.

**Class Imbalance Exacerbates Challenges in Test-Time Adaptation**     Class imbalance exacerbates the challenges of test-time adaptation, as shown in Figure 1 (b). In the HELOC dataset, the source data distribution shows a significant imbalance, with Category 0 accounting for approximately 0.76, leading the model to predict a similarly biased distribution (around 0.82 for Category 0). However, the target data distribution differs, with Category 1 having a higher proportion (approximately 0.58). This indicates that the model relies on the source data distribution for predictions, resulting in biased outputs. Figure 1 (b) highlights how class imbalance causes a mismatch between training and testing distributions, emphasizing the importance of addressing both label distribution shift and class imbalance to develop effective test-time adaptation methods for the tabular domain.

Due to the class imbalance in the source model's predictions, we focus on high-confidence predictions within each class to evaluate its discriminative ability. Specifically, for each class, we select instances with the highest prediction scores (e.g., the top 20% predictions for Class 1) and assess their accuracy. This helps analyze the model's performance on high-confidence predictions. As shown in Figure 1 (c), high-score predictions in the minority class demonstrate relatively high accuracy (0.67 at 20% head labels), but accuracy drops significantly as lower-confidence predictions are included (falling to 0.53 at 100% head labels). This indicates that, despite the impact of class imbalance, high-confidence predictions still exhibit reliable discriminative power.

## 3 METHODOLOGY

In this paper, the proposed algorithm OT3A optimizes label assignment for target domain data, particularly addressing label shift and covariate shift issues. The algorithm consists of three key steps: determination of consistency-confident points, propagation of labels, and correction for covariate shifts. Each step is designed to gradually enhance the label information of target domain data to improve model accuracy in the target domain.

### 3.1 CONSISTENCY-CONFIDENT POINTS

In this step, we determine consistency-confident points (**C2P**) based on the logits of target domain data. These points are then used for subsequent label distribution estimation and covariate shift correction. The determination of high-confidence points relies on two criteria: the maximum difference between the top two logits and the consistency among neighboring points. For

each target domain instance $x_i^t \in D_T^t$, we obtain its logits via the predictive model $f(x_i^t; \theta_t) = [f_1(x_i^t; \theta_t), f_2(x_i^t; \theta_t), \ldots, f_K(x_i^t; \theta_t)]$, where $K$ is the number of classes. Due to the class imbalance, we define high-confidence points based on the quantiles of each predicted class. Specifically, the set of high-confidence points (CFP) is defined as:

$$\mathbf{CFP} = \{x_i^t \mid f_p(x_i^t; \theta_t) \geq \tau_p\}$$

where $p = \arg\max_{j \in Y} f_j(x_i^t; \theta_t)$ and $\tau_p$ is the value corresponding to the $\tau_p$ quantile in the confidence distribution for each class.

To further validate the reliability of high-confidence points, we introduce the concept of consistency points. For each target point $x_i^t$, we consider its neighborhood set $B(x_i^t, D_T^t) = \{x_j^t \mid \|x_i^t - x_j^t\| \leq h_t, x_j^t \in D_T^t\}$, where $\|\cdot, \cdot\|$ is one of distance functions. We define the consistency measure as: $C(x_i^t) = \frac{1}{|B(x_i^t, D_T^t)|} \sum_{j \in B(x_i^t, D_T^t)} \mathbb{I}[\hat{y}_i^t = \hat{y}_j^t]$, where $\hat{y}_j^t$ is the pseudo-label of the neighbor $x_j^t$, and $\mathbb{I}[\cdot]$ is the indicator function, which equals 1 when $\hat{y}_i^r = \hat{y}_j^r$ and 0 otherwise. We define the set of high-consistency points (CSP) as :

$$\mathbf{CSP} = \{x_i^t \mid C(x_i^t) \geq \tau_c\}$$

where $\tau_c$ is the value corresponding to the $\tau_c$ quantile in the consistency distribution. The final set of consistency-confidence points **C2P** consists of points that satisfy both conditions, that is **C2P** = **CFP** $\cap$ **CSP**. **C2P** will be used as the foundation for the subsequent steps.

### 3.2 Label Shift Correction

In Section 4.3, we analyzed the mixture of label shift and covariate shift in tabular data. Existing methods such as ODS fail to accurately estimate the label distribution for each batch, leading to biased model predictions. When label shift occurs continuously, a significant discrepancy remains between the conditional probabilities learned by the model and the true conditional probabilities of test samples.

To address this issue, we adopt the approach proposed by (Berthelot et al., 2019) and calibrate predictions by aligning marginal label distributions. Specifically, we use the ratio of the current label distribution $p_t(y)$ to the original label distribution $p_s(y)$ to weight the model's conditional probability predictions.

Given the model's conditional probability prediction at time $t$, for sample $x_i^t$ as $p_t(y|x_i^t) = \text{softmax}(f(x_i^t; \theta_t))$, the calibrated prediction is calculated as:

$$\tilde{p}_t(y|x_i^t) = \text{norm}\left(p_t(y|x_i^t) \cdot \frac{p_t(y)}{p_s(y)}\right)$$

where $\text{norm}(z)_i = \frac{z_i}{\sum_{i'} z_{i'}}$ is a normalization operation to ensure the adjusted probability distribution sums to 1. Accurately estimating the label distribution $p_t(y)$ during testing remains a critical challenge.

Due to class imbalance in the source domain as the analysis in Section 2.2, the model's conditional probability predictions $p_t(y|x)$ are biased toward the majority class, yet high-confidence samples (e.g., top-ranked minority-class predictions) maintain higher accuracy than the overall $p_t(y)$. Leveraging the fundamental assumption of semi-supervised and unsupervised learning, we propose to optimize predictions using latent structural information. Specifically, we refine the model output by iteratively adjusting predictions, starting from the predictions of highly reliable samples, and constraining both the prediction consistency and feature similarity among samples. To achieve this, we select pseudo-labels from points in C2P for label propagation.

The specific formulas for label propagation are as follows. Two types of affinity matrices are used to model sample relationships, and the details of affinity between samples $i$ and $j$ are as follows:

- $k$-Nearest Neighbor (kNN) Affinity Matrix: $\hat{s}_{ij} = \frac{\mathbb{I}\{x_j \text{ is a k-NN of } x_i\}}{k}$ where $\mathbb{I}\{\cdot\}$ is an indicator function that equals 1 if $f_j$ is among the k nearest neighbors of $f_i$, and 0 otherwise. This matrix encodes feature-space proximity.

- Radial Basis Function (RBF) Affinity Matrix: $\hat{s}_{ij} = \mathbb{I}\{i \neq j\} \exp\left(-\frac{\|x_i - x_j\|^2}{2\sigma^2}\right)$ where $\sigma$ is a variance parameter controlling the Gaussian kernel width. The matrix is row-normalized to ensure probability conservation formulated as $s_{ij} = \frac{\hat{s}_{ij}}{\sum_j \hat{s}_{ij}}$.

The label propagation process is formalized as follows: The initial label matrix $Y$ is defined using one-hot pseudolabels from samples in **C2P**, The label distribution matrix $F$ is then updated iteratively via:

$$F^{(t+1)} = \alpha \cdot SF^{(t)} + (1 - \alpha) \cdot Y$$

where $\alpha \in (0, 1)$ balances the influence of neighboring samples and the initial one-hot pseudolabels from **C2P**. For computational convenience and to establish a neutral starting point, we choose to initialize $F^{(0)} = \mathbf{0}$, the convergence point formula for label propagation is $F^* = (1 - \alpha)(I - \alpha S)^{-1} Y$. Upon convergence yied $F^*$, the class distribution $p_t^L(y)$ is estimated by the latent structural information, here $p_t^L(y) = \frac{1}{|F|} \sum_1^{|F|} \mathbb{I}\{\arg\max(F^*) = y\}$.

Furthermore, to accurately estimate $p_t(y)$, we combine the label distribution $p_t^L(y)$ determined by label propagation with the imbalanced $p_t^I(y) = \frac{1}{B} \sum_i^B p_t(y|x_i^t)$ calculated by the model on the batch. We introduce a dynamic coefficient $\zeta$ to adaptively weight two label distributions. The motivation is that the label distribution within the Consistency–Confidence Point (C2P) set is relatively reliable: its degree of imbalance indicates how likely the true batch-level label distribution deviates from uniformity. Consequently, when the C2P distribution is more imbalanced, we place greater emphasis on label propagation. Formally, let $q$ be the empirical class proportion vector over the C2P set, $q(y) = \frac{1}{|\mathbf{C2P}|} \sum_{x_i \in \mathbf{C2P}} \mathbb{I}\{\arg\max_j p_t(j|x_i) = y\}$. We define an imbalance coefficient $\zeta$ to quantify this discrepancy, $\zeta = \left[\sum_{l=1}^{l=K} \left(q(y) - \frac{1}{K}\right)^2\right]^{\frac{1}{2}}$. In the end, we combined $p_t^L(y)$ and $p_t^I(y)$ to yield the final distribution as $p_t(y) = \zeta * p_t^L(y) + (1 - \zeta) * p_t^I(y)$.

### 3.3 COVARIATE SHIFT CORRECTION

Inspired by the TENT (Wang et al., 2021), we improve classic pseudo-label learning with entropy minimization to address covariate shift. We perform supervised learning on **C2P** to enhance the model's adaptation to target domain data. For each consistency-confident point $x_i^t \in \mathbf{C2P}$, we use its pseudo-label $\hat{y}_i^t$ as the ground truth label for supervised learning: $\mathcal{L}_{\text{pseudo}}(x_i^t) = \mathcal{L}(\hat{y}_i^t, f(x_i^t; \theta_t))$, where $\mathcal{L}(\cdot)$ is the cross-entropy loss function.

Entropy minimization has been a common loss function for the TTA task (Niu et al., 2022; 2023). Intuitively, low-entropy samples contribute more to model adaptation than high-entropy samples. Given that points in **C2P** correspond to high-confidence samples, which are equivalent to low-entropy samples. Our objective is to minimize the entropy of the predicted distribution for these points, encouraging the model to make more confident predictions on them, thus, we employ entropy minimization on **C2P** as follows: $\mathcal{L}_{\text{entropy}}(x_i^t) = -\sum_{c=1}^K f_c(x_i^t; \theta_t) \log(f_c(x_i^t; \theta_t))$, where $f_c(x_i^t; \theta_t)$ is the predicted probability of class $c$ for target domain instance $x_i^t$. Therefore, we can finally update $\theta_t$ by:

$$\min_{\theta_t} E_{x_i^t \in \mathbf{C2P}} \mathcal{L}_{\text{pseudo}}(x_i^t) + \mathcal{L}_{\text{entropy}}(x_i^t).$$

## 4 EXPERIMENTS AND RESULTS

### 4.1 EXPERIMENTAL SETUP

**Datasets and Evaluation.** In our implementation, we follow the online test-time adaptation setting, where the source model is trained on training data and adapted to shifted test data without any access to the source training data. Specifically, we train the source model on source data and select the best model based on the validation set. Then, our OT3A approach and existing TTA methods are evaluated on the shifted test set. To demonstrate our approach across various test-time distribution shifts, we test our method under six natural distribution shifts datasets – HELOC, Voting, Diabetes, ASSISTments, Hospital Readmission, and Childhood Lead in TableShift benchmark (Gardner et al., 2023). As shown in Figure 1 (b), tabular data often exhibit extreme class imbalance. Since accuracy

may not be effective in these cases, we use macro F1 score (F1) and balanced accuracy (bAcc.) as the primary evaluation metrics.

**Baselines.** We compare OT3A with 6 baselines—PL (Lee, 2013) updates entire networks by minimizing the cross-entropy between prediction and the pseudo label; TTT (Sun et al., 2020a) mitigate deterioration of test-time adaptation performance through feature alignment strategies; TENT (Wang et al., 2021) updates the model parameters with entropy minimization loss; EATA (Niu et al., 2022) performs activate sample selection for adaptation and Fisher regularization for anti-forgetting to achieve strong predicting performance; SAR (Niu et al., 2023) conducts sample filtering based on test entropy and update model parameters to a flat minimum to achieve well and robust performance; ODS (Zhou et al., 2023) decouples the mixed distribution shift and then addresses covariate and label distribution shifts accordingly.

## 4.2 MAIN RESULTS

Table 7 summarizes the experimental results using MLP as the backbone network. Figure 2 illustrates the performance improvement of the proposed method across different backbone networks (MLP and Transformer). Figure 3 shows the estimation of the label distribution by our method in each batch. All reported values are the mean and standard error of three repeated experiments. More experimental results and details are in the Supplemental Material.

Figure 2: The average performance of OT3A approach and comparison methods using MLP & Tab-Transformer.

**OT3A outperforms existing TTA methods.** To evaluate the effectiveness of OT3A, we report the detailed experimental results using MLP as backbone model in Table 7. Across a diverse set of six datasets and measured by both balanced accuracy and Marco F1 score, OT3A consistently achieves the best performance, often by a significant margin, compared to a model without adaptation (Source) and several existing state-of-the-art TTA methods. This strongly supports the claim that OT3A is a superior method for addressing distribution shifts in tabular data.

**OT3A stably improves model performance under shifts.** Figure 2 clearly illustrates that our proposed OT3A method achieves substantial performance improvements over other Test-Time Adaptation (TTA) approaches across two backbone network architectures: MLP and Transformer. Specifically, for the MLP architecture, OT3A delivers an approximate 7 percentage point gain in balanced accuracy, increasing from around 59% to approximately 66% compared to the "Source" baseline. Similarly, for the Transformer architecture, OT3A achieves a comparable improvement, with balanced accuracy rising from roughly 58% to around 66%. In contrast, other TTA methods such as PL, TTT, TENT, EATA, LAME, ODS, and SAR show minimal improvements. This clearly demonstrates that regardless of the underlying model architecture, OT3A method consistently enhances the performance in tackling tabular test time shift problem.

**OT3A achieves accurate label distribution correction.** We adopt KL-divergence (Kullback & Leibler, 1951) to measure the error between the ground-truth label distribution and the estimated label distribution at each timestamp t. While ODS (Zhou et al., 2023) attempts to address both covariate and label distribution shifts, it underperforms OT3A in most scenarios. A key reason for this performance difference is OT3A's superior ability to accurately estimate the shifted label distribution $p_t(y)$ over time. As demonstrated in Figure 3, OT3A achieves a significantly lower KL-divergence compared to ODS at each timestamp, indicating a more precise estimation of the true label distribution.

Table 1: Performance of the OT3A approach and compared baselines on 6 datasets using MLP. The best performance is in **Bold**. More experimental results are in the Supplemental Material.

| Method | HELOC | | Voting | | Diabetes | |
|---|---|---|---|---|---|---|
| | bAcc | F1 | bAcc | F1 | bAcc | F1 |
| Source | 53.25 ±3.5 | 40.02 ±5.3 | 75.66 ±0.4 | 77.24 ±0.2 | 55.22 ±0.1 | 55.50 ±0.0 |
| PL | 51.82 ±1.3 | 34.92 ±2.3 | 75.61 ±0.3 | 76.63 ±0.5 | 55.10 ±0.1 | 55.30 ±0.1 |
| TTT | 53.20 ±1.5 | 38.21 ±3.6 | 76.80 ±0.5 | 77.64 ±0.2 | 55.41 ±0.2 | 55.73 ±0.1 |
| TENT | 54.24 ±5.8 | 39.91 ±2.6 | 74.09 ±0.6 | 74.76 ±0.3 | 55.00 ±0.0 | 55.00 ±0.0 |
| EATA | 54.25 ±3.6 | 40.02 ±1.6 | 76.20 ±0.5 | 77.79 ±0.1 | 55.20 ±0.4 | 55.50 ±0.1 |
| LAME | 50.00 ±0.0 | 30.10 ±0.6 | 54.60 ±0.4 | 46.80 ±0.1 | 54.82 ±0.2 | 54.80 ±0.6 |
| ODS | 50.00 ±0.0 | 30.10 ±0.0 | 54.60 ±0.5 | 46.80 ±0.0 | 54.80 ±0.1 | 54.80 ±0.3 |
| SAR | 54.74 ±0.9 | 33.16 ±2.9 | 64.20 ±0.5 | 59.79 ±0.1 | 53.48 ±0.4 | 54.81 ±0.9 |
| **OT3A** | **64.12** ±0.9 | **63.97** ±0.7 | **78.32** ±0.5 | **78.97** ±0.2 | **71.29** ±0.5 | **67.87** ±0.2 |

| Method | ASSISTments | | Hospital Readmission | | Childhood Lead | |
|---|---|---|---|---|---|---|
| | bAcc | F1 | bAcc | F1 | bAcc | F1 |
| Source | 60.81 ±3.3 | 46.42 ±1.8 | 60.59 ±0.2 | 59.12 ±1.6 | 50.00 ±0.0 | 47.90 ±0.0 |
| PL | 57.30 ±0.3 | 44.49 ±0.5 | 60.65 ±0.0 | 59.16 ±0.6 | 50.00 ±0.0 | 47.90 ±0.0 |
| TTT | 60.02 ±0.6 | 45.92 ±0.2 | 60.46 ±0.0 | 59.62 ±0.1 | 50.00 ±0.0 | 47.90 ±0.0 |
| TENT | 56.41 ±0.3 | 63.99 ±0.2 | 60.15 ±0.3 | 53.75 ±1.0 | 50.00 ±0.0 | 47.90 ±0.0 |
| EATA | 60.81 ±0.2 | 46.42 ±0.1 | 60.16 ±0.3 | 53.68 ±0.9 | 50.00 ±0.0 | 47.90 ±0.0 |
| LAME | 51.30 ±0.1 | 41.40 ±0.1 | 54.90 ±0.3 | 46.69 ±1.4 | 50.00 ±0.0 | 47.90 ±0.0 |
| ODS | 51.30 ±0.1 | 41.40 ±0.1 | 54.90 ±0.3 | 46.69 ±1.4 | 50.00 ±0.0 | 47.90 ±0.0 |
| SAR | 60.81 ±0.1 | 46.42 ±0.1 | 57.19 ±0.3 | 51.98 ±0.9 | 50.00 ±0.0 | 47.90 ±0.0 |
| **OT3A** | **63.97** ±1.1 | **61.89** ±0.6 | **61.03** ±0.2 | **59.92** ±0.9 | **61.29** ±0.1 | **66.79** ±0.2 |

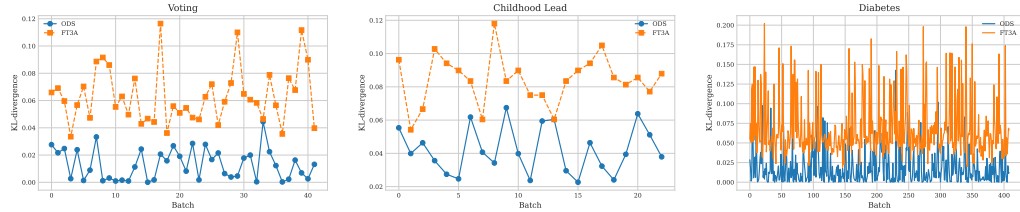

Figure 3: The KL-divergence between the estimated and ground-truth label distributions for ODS and OT3A at each batch during test-time adaptation.

### 4.3 ABLATION ANALYSIS

**Parameter sensitivity** We analyze the parameter sensitivity of our method as Figure 4, which illustrates the effect of varying $\tau_p$, $\tau_c$, batch size, and learning rate on performance, measured by Balanced Accuracy and F1 Score. Specifically, we focus on: (1) the parameters $\tau_p$ and $\tau_c$, which determine the selection of **C2P**; (2) the model's training hyperparameters, batch size and learning rate; (3) parameters related to the label propagation, include different affinity matrix and weighting factor for label propagation $\alpha$. OT3A demonstrates a degree of robustness to parameter selection. The performance is not highly sensitive to changes in $\tau_p$, $\tau_c$, and batch size within the tested ranges. The learning rate has a more pronounced effect, highlighting its importance in the optimization process.

**Robustness of affinity matrices.** Table 2 further analyzes the introduced affinity matrices for performing output adaptation, which is RBF affinity with parameter $\sigma$ and kNN affinity with parameter $k$ with their weighting factor $\alpha$. We conduct experiments to predict the HELOC dataset using adjacency matrices constructed with both the Radial Basis Function (RBF) with $\sigma \in \{0.1, 1.0, 2.5, 5.0\}$ and k-Nearest Neighbors (kNN) with $k \in \{3, 5, 10, 15\}$, under various settings of the weighting factor $\alpha \in \{0.2, 0.5, 0.8\}$. And the results are demonstrated in Table 2. We observe that, for RBF affinity, the results remain largely consistent when $\sigma > 0.1$. This stability arises because, with a large $\sigma$ value, the RBF matrix behaves similarly to a uniform dense matrix, leading to similar results. Meanwhile, when we set $\sigma = 0.1$, the RBF matrix becomes notably sparse, making it less

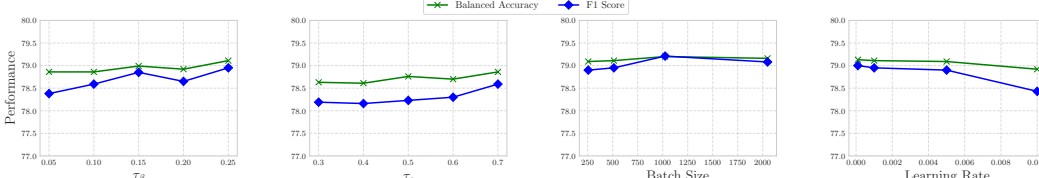

Figure 4: Robustness of batch size and hyperparameters on the Voting dataset using the MLP backbone model. The results indicate that minor perturbations to the hyperparameters of OT3A do not significantly affect its performance, demonstrating the practical robustness of OT3A.

Table 2: Performance of different affinity matrices (RBF affinity $\sigma$ and kNN affinity $k$) and weighting factors $\alpha$ on the HELOC dataset. The gray row color indicates the largest impact parameters.

|  | $\alpha = 0.2$ | | $\alpha = 0.5$ | | $\alpha = 0.8$ | |
|---|---|---|---|---|---|---|
|  | bAcc | F1 | bAcc | F1 | bAcc | F1 |
| $\sigma = 0.1$ | 60.96 $\pm$0.1 | 60.07 $\pm$0.1 | 60.20 $\pm$0.1 | 58.89 $\pm$0.1 | 57.62 $\pm$0.1 | 54.11 $\pm$0.2 |
| $\sigma = 0.5$ | 63.41 $\pm$0.0 | 63.50 $\pm$0.0 | 63.41 $\pm$0.0 | 63.51 $\pm$0.0 | 63.25 $\pm$0.1 | 63.32 $\pm$0.1 |
| $\sigma = 1.0$ | 63.46 $\pm$0.1 | 63.57 $\pm$0.1 | 63.28 $\pm$0.0 | 63.37 $\pm$0.0 | 63.39 $\pm$0.0 | 63.48 $\pm$0.0 |
| $\sigma = 2.5$ | 63.57 $\pm$0.0 | 63.66 $\pm$0.0 | 63.51 $\pm$0.1 | 63.60 $\pm$0.1 | 63.53 $\pm$0.1 | 63.63 $\pm$0.1 |
| $\sigma = 5.0$ | 63.66 $\pm$0.0 | 63.73 $\pm$0.0 | 63.56 $\pm$0.1 | 63.64 $\pm$0.1 | 63.53 $\pm$0.1 | 63.62 $\pm$0.1 |
| $k = 3$ | 63.56 $\pm$0.1 | 63.61 $\pm$0.0 | 63.50 $\pm$0.2 | 63.61 $\pm$0.2 | 63.21 $\pm$0.1 | 63.30 $\pm$0.1 |
| $k = 5$ | 63.37 $\pm$0.1 | 63.46 $\pm$0.1 | 63.54 $\pm$0.2 | 63.64 $\pm$0.2 | 63.27 $\pm$0.2 | 63.37 $\pm$0.1 |
| $k = 10$ | 63.43 $\pm$0.3 | 63.52 $\pm$0.3 | 63.55 $\pm$0.1 | 63.65 $\pm$0.1 | 63.53 $\pm$0.2 | 63.47 $\pm$0.0 |
| $k = 15$ | 63.33 $\pm$0.2 | 63.42 $\pm$0.3 | 63.50 $\pm$0.3 | 63.74 $\pm$0.1 | 63.47 $\pm$0.1 | 63.56 $\pm$0.1 |

suitable for leveraging latent structure information. The results demonstrate that OT3A is robust to slight changes in all hyperparameters.

**Effect of each component** The ablation study, detailed in Table 3, provides critical insights into the contribution of individual components within our proposed OT3A framework. Across a diverse set of tabular datasets, the removal of any core module consistently resulted in a degradation of performance, as measured by both balanced accuracy (bAcc) and F1 score. This underscores the integral role each component plays in the overall effectiveness of OT3A in mitigating test-time domain shift. Notably, the absence of Label Shift Correction (LSC) mechanisms (w/o CFP, w/o CSP) generally led to a more pronounced decline in bAcc. This highlights the importance of aligning the model's class predictions with the target domain's distribution, particularly in scenarios with potential class imbalance. Conversely, the exclusion of Covariate Shift Correction (CSC) strategies (w/o EM, w/o PL) typically resulted in a more significant reduction in the F1 score, indicating the necessity of adapting the feature representations learned from the source domain to better suit the characteristics of the target data, thereby maintaining a balance between precision and recall. The synergistic effect of these components is further evidenced by the fact that the complete OT3A method consistently outperforms any of its ablated variants. These findings collectively validate the design choices within OT3A, demonstrating that both label and covariate shift correction are crucial for achieving robust and effective test-time adaptation on tabular data.

## 5 RELATED WORK

Test-Time Adaptation. Machine learning models often suffer performance drops when confronted with distribution shifts between training and test data (Wang et al., 2024; Gong et al., 2012; Ganin et al., 2016; Tzeng et al., 2017; Saito et al., 2020; Lee et al., 2019; Xiao et al., 2023). Online Test-Time Adaptation (OTTA) addresses this challenge by adapting a pre-trained model to new, unseen test distributions on the fly, using only unlabeled test inputs during inference (Azizzadenesheli et al., 2019; Park et al., 2023; Alexandari et al., 2020). Pioneering OTTA methods in computer vision predominantly tackle covariate shifts–changes in the input feature distribution $p_T(x)$ while the label conditional $p(y|x)$ remains stable (Cao et al., 2019; Cui et al., 2019; Kang et al., 2020; Hong et al., 2021). To mitigate these covariate shifts, early approaches update model parameters or normaliza-

Table 3: Ablation study of our OT3A approach with its variants. "LSC" indicates Label Shift Correction in Section 3.2 and "CSC" indicates Covariant Shift Correction in Section 3.3. "CFP" and "CSP" belong to consistency-confident points "C2P" in Section 3.1. The gray row color indicates the largest impact components.

| Method | Ablation | | | | HELOC | | Voting | | Diabetes | |
|---|---|---|---|---|---|---|---|---|---|---|
| | CFP | CSP | EM | PL | bAcc | F1 | bAcc | F1 | bAcc | F1 |
| **OT3A** | ✔ | ✔ | ✔ | ✔ | **64.12** $_{\pm0.9}$ | **63.97** $_{\pm0.7}$ | **78.32** $_{\pm0.5}$ | **78.97** $_{\pm0.2}$ | **71.29** $_{\pm0.5}$ | **67.87** $_{\pm0.2}$ |
| – w/o LSC | ✗ | ✗ | ✔ | ✔ | 57.78 $_{\pm0.3}$ | 50.03 $_{\pm0.1}$ | 74.87 $_{\pm0.2}$ | 75.84 $_{\pm0.2}$ | 56.57 $_{\pm0.1}$ | 57.55 $_{\pm0.0}$ |
| | ✗ | ✔ | ✔ | ✔ | 63.64 $_{\pm0.5}$ | 62.35 $_{\pm0.6}$ | 76.16 $_{\pm0.5}$ | 77.14 $_{\pm0.3}$ | 66.25 $_{\pm0.1}$ | 64.21 $_{\pm0.1}$ |
| | ✔ | ✗ | ✔ | ✔ | 63.87 $_{\pm0.8}$ | 63.87 $_{\pm0.6}$ | 78.51 $_{\pm0.6}$ | 79.09 $_{\pm0.2}$ | 69.92 $_{\pm0.2}$ | 67.89 $_{\pm0.1}$ |
| – w/o CSC | ✔ | ✔ | ✗ | ✗ | 64.07 $_{\pm0.3}$ | 63.97 $_{\pm0.3}$ | 77.87 $_{\pm0.4}$ | 78.59 $_{\pm0.1}$ | 69.71 $_{\pm0.0}$ | 67.82 $_{\pm0.1}$ |
| | ✔ | ✔ | ✗ | ✔ | 63.97 $_{\pm0.4}$ | 63.86 $_{\pm0.3}$ | 77.88 $_{\pm0.5}$ | 78.59 $_{\pm0.1}$ | 69.97 $_{\pm0.3}$ | 67.82 $_{\pm0.2}$ |
| | ✔ | ✔ | ✔ | ✗ | 64.14 $_{\pm0.1}$ | 64.06 $_{\pm0.1}$ | 77.86 $_{\pm0.5}$ | 78.57 $_{\pm0.4}$ | 69.90 $_{\pm0.1}$ | 67.89 $_{\pm0.1}$ |
| Source | - | - | - | - | 53.25 $_{\pm3.5}$ | 40.02 $_{\pm5.3}$ | 75.66 $_{\pm0.4}$ | 77.24 $_{\pm0.2}$ | 55.22 $_{\pm0.1}$ | 55.50 $_{\pm0.0}$ |

| Method | Ablation | | | | ASSISTments | | Hospital Readmission | | Childhood Lead | |
|---|---|---|---|---|---|---|---|---|---|---|
| | CFP | CSP | EM | PL | bAcc | F1 | bAcc | F1 | bAcc | F1 |
| **OT3A** | ✔ | ✔ | ✔ | ✔ | **63.97** $_{\pm1.1}$ | **61.89** $_{\pm0.6}$ | **61.03** $_{\pm0.2}$ | **59.92** $_{\pm0.9}$ | **61.58** $_{\pm0.1}$ | **67.18** $_{\pm0.2}$ |
| – w/o LSC | ✗ | ✗ | ✔ | ✔ | 63.04 $_{\pm0.7}$ | 54.53 $_{\pm0.5}$ | 60.41 $_{\pm0.4}$ | 59.10 $_{\pm0.2}$ | 50.00 $_{\pm0.0}$ | 47.90 $_{\pm0.0}$ |
| | ✗ | ✔ | ✔ | ✔ | 59.54 $_{\pm0.6}$ | 48.87 $_{\pm0.4}$ | 57.21 $_{\pm0.4}$ | 51.65 $_{\pm0.3}$ | 61.45 $_{\pm0.4}$ | 67.01 $_{\pm0.3}$ |
| | ✔ | ✗ | ✔ | ✔ | 63.67 $_{\pm0.5}$ | 61.93 $_{\pm0.4}$ | 58.78 $_{\pm0.4}$ | 55.46 $_{\pm0.3}$ | 61.45 $_{\pm0.1}$ | 67.01 $_{\pm0.1}$ |
| – w/o CSC | ✔ | ✔ | ✗ | ✗ | 62.65 $_{\pm0.4}$ | 55.94 $_{\pm0.3}$ | 56.21 $_{\pm0.7}$ | 49.37 $_{\pm0.2}$ | 61.48 $_{\pm0.1}$ | 66.94 $_{\pm0.3}$ |
| | ✔ | ✔ | ✗ | ✔ | 63.20 $_{\pm0.5}$ | 61.71 $_{\pm0.4}$ | 56.62 $_{\pm0.3}$ | 50.26 $_{\pm0.1}$ | 61.35 $_{\pm0.3}$ | 66.86 $_{\pm0.5}$ |
| | ✔ | ✔ | ✔ | ✗ | 63.69 $_{\pm0.8}$ | 62.12 $_{\pm0.3}$ | 56.44 $_{\pm0.3}$ | 49.86 $_{\pm0.4}$ | 61.29 $_{\pm0.2}$ | 66.94 $_{\pm0.3}$ |
| Source | - | - | - | - | 60.81 $_{\pm3.3}$ | 46.42 $_{\pm1.8}$ | 60.59 $_{\pm0.2}$ | 59.12 $_{\pm1.6}$ | 50.00 $_{\pm0.0}$ | 47.90 $_{\pm0.0}$ |

tion statistics at test time. For example, TENT adapts a model by updating batch normalization parameters via entropy minimization on each test batch (Wang et al., 2021). TTT optimizes a self-supervised auxiliary task (*e.g.*, predicting image rotations) on each unlabeled test sample to improve robustness under shifts (Sun et al., 2020a). More generally, prior test-time adaptation techniques employ strategies like recalibrating batch norm statistics, minimizing prediction entropy, or performing self-supervised learning on test data (Schneider et al., 2020; Lim et al., 2023; Gong et al., 2022). Several methods also incorporate pseudo-labeling, treating high-confidence model predictions on unlabeled test examples as surrogate labels to further adapt the model (Niu et al., 2022; Wang et al., 2021; Lee, 2013). Such optimization-based OTTA methods have proven effective for covariate shifts, especially in vision tasks, and they can significantly improve robustness without requiring any source data during inference. By contrast, OTTA for tabular data has been relatively underexplored and poses distinct challenges. Tabular datasets are ubiquitous in real-world domains such as finance and healthcare, with heterogeneous feature types and no clear spatial or sequential structure (Ren et al., 2024; Fang et al., 2024; Bahri et al., 2022). Methods that work well for test-time adaptation in computer vision often fail to transfer to tabular tasks, which is a key motivation for our work. Crucially, OOD issues in tabular data are pervasive in industry, yet effective adaptation solutions remain scarce.

# 6 CONCLUSION AND LIMITATION

In this paper, we address the challenge of OTTA for tabular data. This is a significant and highly challenging problem, particularly due to the presence of class imbalance in tabular data and the mixture of multi-types of shifts. Our OT3A method identifies a subset of high-confidence pseudo-labeled samples to correct label shift and further utilizes entropy minimization and pseudo-label learning techniques to address covariate shift. Experimental results demonstrate that OT3A significantly outperforms existing adaptation methods. Besides, our method also has some limitations. It is constrained to extend this method to tasks beyond classification. Since OT3A relies on the generation of pseudo-labels, adapting it to continuous predictions (e.g., regression tasks) is non-trivial. Note that other methods also face similar challenges, since estimating prediction entropy for continuous outputs remains difficult.

## ETHICS STATEMENT

Our research focuses on the fundamental machine learning challenge of domain generalization and test-time adaptation for tabular data. We primarily utilized publicly available benchmark datasets, and our work does not involve personally identifiable or sensitive data. During the preparation of this manuscript, we employed large language models (e.g., GPT) for language polishing and grammatical correction to enhance readability.

## REPRODUCIBILITY STATEMENT

To ensure the reproducibility of our work, we have made our code and experimental setup publicly available. The full source code for our proposed method, OT3A, along with the scripts used to conduct all experiments reported in this paper, will be included in the supplementary materials and released publicly upon publication. Our paper provides detailed descriptions of the model architecture, training procedures, and the specific distribution shift scenarios created for evaluation (see Sections 4.) The Appendix B further contains details on hyperparameter settings for all compared methods, ensuring that our results can be fully replicated.

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

## A    Detailed Experimental Setup

We have briefly introduced experimental setup in our main manuscript. Here, we provide detailed experimental setup in this section.

Table 4: Summary of the datasets used in our experiments, including the total number of instances (Total Instances), the number of instances allocated to training, validation, and test sets (Training Set, Validation Set, Test Set), the total number of features (Total Features), and a breakdown into numerical and categorical features (Numerical Features, Categorical Features). All tasks involve binary classification.

| Statistic | HELOC | Voting | Hospital Readmission | ASSISTments | Childhood Lead | Diabetes |
|---|---|---|---|---|---|---|
| Total Instances | 9,412 | 60,376 | 89,542 | 24,00998 | 24,749 | 1,299,758 |
| Training Set | 2,220 | 34,796 | 34,288 | 21,3256 | 11,807 | 969,229 |
| Validation Set | 278 | 4,349 | 4,286 | 26,6566 | 1,476 | 121,154 |
| Test Set | 6,914 | 21,231 | 50,968 | 1,906 | 11,466 | 209,375 |
| Total Features | 22 | 54 | 46 | 26 | 7 | 25 |
| Numerical Features | 20 | 8 | 12 | 9 | 4 | 6 |
| Categorical Features | 2 | 46 | 34 | 17 | 3 | 19 |

## A.1 DATASETS

In our experiments, we verify our method across six different datasets—HELOC, Voting, Hospital Readmission, ASSISTmments, Childhood Lead, and Diabetes—within the Tableshift Benchmark, all of which include natural distribution shifts between training and test data. We give the details of each dataset as follows and detailed statistics specifications of each dataset are listed in Table 4.

- HELOC: Home Equity Line of Credit dataset is the dataset to predict whether the applicant will repay their HELOC account within two years; which is a line of credit typically offered by a bank as a percentage of home equity. Data is split with respect to external risk estimation value; lower ones are used for test data.

- Voting: American National Election Studies provide classification task of U.S. presidential election participation. Domain shift is given by the geographic region of surveyees.

- Hospital Readmisson: Diabetes Readmission represents ten years (1999-2008) of clinical care at 130 US hospitals and integrated delivery networks. Each row concerns hospital records of patients diagnosed with diabetes, who underwent laboratory, medications, and stayed up to 14 days. The goal is to determine the early readmission of the patient within 30 days of discharge. Admission sources are different between train and test data.

- Childhood Lead: This task predicts elevated blood lead levels in children using NHANES data, with 27,499 observations. A distribution shift is introduced by splitting the data based on poverty using the poverty-income ratio (PIR) as a threshold. Those with a PIR of 1.3 or lower are held out for testing, simulating risk assessment in lower-income households.

- Diabetes: This task predicts diabetes using BRFSS data, focusing on racial shifts across 1.4 million observations. Distribution shift occurs by focusing on the differences in diabetes risk between racial and ethnic groups, particularly highlighting the higher risk faced by non-white groups compared to White non-Hispanic individuals.

- ASSISTments. ASSISTMENTS dataset in education field. The ASSISTments tutoring platform is a free, web-based, data-driven tutoring platform for students in grades 3-12. ASSISTMENTS dataset contains affect predictions such as such as boredom, confusion, frustration, and engaged problem-solving behavior on students who use the ASSISTMENTS tutoring platform.

## A.2 BASELINE DETAILS

To compare with our method OT3A, we have selected the following Test-Time Adaptation (TTA) methods. Detailed descriptions of these methods are provided below.

- PL: Pseudo-Labeling (PL) leverages a pseudo-labeling strategy to update model parameters during test time.

- TTT: Test-time training adapts to a new test distribution on the fly by optimizing a model for each test input using self-supervision.

- TENT: Test Entropy minimization (TENT) updates the scale and bias parameters in the batch normalization layer during test time by minimizing entropy within a given test batch.

- EATA: Efficient Anti-forgetting Test-time Adaptation (EATA) mitigates the risk of unreliable gradients by filtering out high-entropy samples and applying a Fisher regularizer to constrain key model parameters during adaptation.

- SAR: Sharpness-Aware and Reliable optimization (SAR) builds on TENT by filtering samples with large entropy, which can cause model collapse during test time, using a predefined threshold.

- LAME: Laplacian Adjusted Maximum-likelihood Estimation (LAME) employs an output adaptation strategy during test-time, focusing on adjusting the model's output probabilities rather than tuning its parameters.

- ODS: An algorithm that decouples the mixed distribution shift and then separately addresses covariate and label distribution shifts.

## B  IMPLEMENTATION DETAILS

In this section, we provide the details of backbone model, configuration of training and testing phase to enhance the reporducibility. All experiments are conducted on a Linux server with one NVIDIA GeForce RTX 3050Ti GPU.

**Backbone Models.**  For all experiments, we use two representative deep tabular models: MLP, Tabtransformer as the backbone model.

- MLP: Multi-Layer Perceptron (MLP) is a foundational deep learning architecture characterized by multiple layers of interconnected nodes, where each node applies a non-linear activation function to a weighted sum of its inputs. In the tabular domain, MLP is often employed as a default deep learning model, with each input feature corresponding to a node in the input layer.

- TabTransformer: TabTransformer is a deep learning architecture specifically designed for processing tabular data. It leverages the power of transformer-based attention mechanisms to model complex interactions between categorical and numerical features. The model encodes categorical data using embedding layers and applies self-attention to capture feature dependencies, providing an alternative to traditional approaches like MLP for tabular data tasks.

### B.1  HYPERPARAMETERS FOR TRAINING PHASE.

For training the source model, we follow the TableShift benchmark for all setting of training hyperparameters. Specifically, we train each backbone model with a batch size of 512 for several epochs, depending on the model's convergence as evaluated on the validation set. The AdamW optimizer is used with a learning rate of 0.01 and a weight decay of 0.01.

### B.2  HYPERPARAMETERS FOR TTA BASELINES

In this subsection, we present the parameter settings for baseline methods. PL, TENT , and SAR require three main hyperparameters: the learning rate, the number of adaptation steps per batch, and the option for episodic adaptation (i.e., resetting the model after processing each batch of data). Specifically, PL and TENT were configured with a learning rate of 0.0001, 1 adaptation step, and episodic updates. Additionally, SAR requires a threshold to filter high-entropy samples, with the learning rate set to 0.001, 1 adaptation step, and episodic updates.

For the TTT method, a proxy task needs to be defined during training. We adopted a self-supervised proxy task similar to VIME, in which 15% of values in each column are randomly replaced with other values from the same column. Then, an MLP is used as a discriminator network to identify the replaced values and recover them.

Table 5: The Notation for the parameters in the OT3A method and the parameter settings used in the main experiments.

| Notation | Meaning | Value |
|---|---|---|
| $\tau_p$ | The quantile in the confident distribution | 0.25 |
| $\tau_c$ | The quantile in the consistent distribution | 0.7 |
| $\alpha$ | Weighting factor for label propagation | 1.0 |
| $\sigma$ | Variance parameter in RBF Affinity Matrix | 1.0 |
| $k$ | k-Nearest Neighbor in KNN Affinity Matrix | - |

Table 6: Performance of the OT3A approach and compared baselines on 6 datasets using MLP. The best performance is in **Bold**. More experimental results are in the Supplemental Material.

| Method | HELOC | | | Voting | | | Diabetes | | |
|---|---|---|---|---|---|---|---|---|---|
| | Acc | bAcc | F1 | Acc | bAcc | F1 | Acc | bAcc | F1 |
| Source | 53.21 ±4.3 | 53.25 ±3.5 | 40.02 ±5.3 | 78.69 ±0.3 | 75.66 ±0.4 | 77.24 ±0.2 | 83.32 ±0.2 | 55.22 ±0.1 | 55.50 ±0.0 |
| PL | 54.78 ±1.1 | 51.82 ±1.3 | 34.92 ±2.3 | 75.87 ±0.4 | 75.61 ±0.3 | 76.63 ±0.5 | **83.81** ±0.3 | 55.10 ±0.1 | 55.30 ±0.1 |
| TTT | 53.05 ±3.1 | 53.20 ±1.5 | 38.21 ±3.6 | 76.08 ±0.5 | 76.80 ±0.5 | 77.64 ±0.2 | 83.29 ±0.0 | 55.41 ±0.2 | 55.73 ±0.1 |
| TENT | 54.35 ±2.3 | 54.24 ±5.8 | 39.91 ±2.6 | 78.07 ±0.6 | 74.09 ±0.6 | 74.76 ±0.3 | 83.32 ±0.0 | 55.00 ±0.0 | 55.00 ±0.0 |
| EATA | 54.37 ±2.1 | 54.25 ±3.6 | 40.02 ±1.6 | 78.13 ±0.3 | 76.20 ±0.3 | 77.79 ±0.1 | 83.50 ±0.3 | 55.20 ±0.4 | 55.50 ±0.1 |
| LAME | 43.10 ±4.6 | 50.00 ±0.0 | 30.10 ±0.6 | 63.50 ±2.1 | 54.60 ±0.4 | 46.80 ±0.1 | 83.24 ±0.1 | 54.82 ±0.2 | 54.80 ±0.6 |
| ODS | 43.10 ±4.6 | 50.00 ±0.0 | 30.10 ±0.0 | 63.50 ±2.1 | 54.60 ±0.5 | 46.80 ±0.0 | 83.24 ±0.1 | 54.80 ±0.1 | 54.80 ±0.3 |
| SAR | 52.32 ±2.6 | 54.74 ±0.9 | 33.16 ±2.9 | 78.13 ±0.6 | 64.20 ±0.5 | 59.79 ±0.1 | 82.98 ±0.2 | 53.48 ±0.4 | 54.81 ±0.9 |
| **OT3A** | **64.56** ±1.9 | **64.12** ±0.9 | **63.97** ±0.7 | **80.21** ±0.4 | **78.32** ±0.5 | **78.97** ±0.2 | 79.91 ±0.2 | **71.29** ±0.5 | **67.87** ±0.2 |

| Method | ASSISTments | | | Hospital Readmission | | | Childhood Lead | | |
|---|---|---|---|---|---|---|---|---|---|
| | Acc | bAcc | F1 | Acc | bAcc | F1 | Acc | bAcc | F1 |
| Source | 51.57 ±3.2 | 60.81 ±3.3 | 46.42 ±1.8 | 60.65 ±0.3 | 60.59 ±0.2 | 59.12 ±1.6 | 91.93 ±0.0 | 50.00 ±0.0 | 47.90 ±0.0 |
| PL | 56.45 ±0.4 | 57.30 ±0.3 | 44.49 ±0.5 | 60.27 ±0.2 | 60.65 ±0.0 | 59.16 ±0.6 | 91.93 ±0.0 | 50.00 ±0.0 | 47.90 ±0.0 |
| TTT | 55.86 ±1.3 | 60.02 ±0.6 | 45.92 ±0.2 | 61.02 ±0.3 | 60.46 ±0.0 | 59.62 ±0.1 | 91.93 ±0.0 | 50.00 ±0.0 | 47.90 ±0.0 |
| TENT | 50.87 ±0.3 | 56.41 ±0.3 | 63.99 ±0.2 | 61.34 ±0.3 | 60.15 ±0.3 | 53.75 ±1.0 | 91.93 ±0.0 | 50.00 ±0.0 | 47.90 ±0.0 |
| EATA | 55.86 ±0.2 | 60.81 ±0.2 | 46.42 ±0.1 | 61.36 ±0.3 | 60.16 ±0.3 | 53.68 ±0.9 | 91.93 ±0.0 | 50.00 ±0.0 | 47.90 ±0.0 |
| LAME | 45.12 ±0.2 | 51.30 ±0.1 | 41.40 ±0.1 | 61.39 ±0.1 | 54.90 ±0.3 | 46.69 ±1.4 | 91.93 ±0.0 | 50.00 ±0.0 | 47.90 ±0.0 |
| ODS | 45.12 ±0.2 | 51.30 ±0.1 | 41.40 ±0.1 | 61.39 ±0.1 | 54.90 ±0.3 | 46.69 ±1.4 | 91.93 ±0.0 | 50.00 ±0.0 | 47.90 ±0.0 |
| SAR | 55.86 ±0.2 | 60.81 ±0.1 | 46.42 ±0.1 | 61.38 ±0.2 | 57.19 ±0.3 | 51.98 ±0.9 | 91.93 ±0.0 | 50.00 ±0.0 | 47.90 ±0.0 |
| **OT3A** | **62.28** ±0.4 | **63.97** ±1.1 | **61.89** ±0.6 | 61.03 ±0.2 | **61.03** ±0.2 | **59.92** ±0.9 | **93.76** ±0.0 | **61.29** ±0.1 | **66.79** ±0.2 |

For TTT, EATA and LAME , we followed the original authors' hyperparameter settings, except for the learning rate and adaptation steps. TTT and EATA were configured with a learning rate of 0.00001, 10 adaptation steps, and episodic updates. LAME and ODS only adjust output logits and therefore do not require hyperparameters related to gradient updates.

### B.3 HYPERPARAMETERS FOR OT3A

In Section 4.3, we analyze the impact of various parameters in the OT3A method. The parameter settings for our main experiments are listed in Table 5. From the ablation studies, we observe that our experiments exhibit considerable robustness to most parameters within a certain range.

## C ADDITIONAL EXPERIMENTS

In Section 4, we omitted the presentation of the Acc(accuracy) metric. Here, we supplement the Acc results from our experiments and additionally provide the baseline results using TabTransformer.

## D LIMITATIONS AND BROADER IMPACTS

### D.1 LIMITATIONS

Our OT3A method can identify high-confidence pseudo-labeled subsets to mitigate label shift and further leverage entropy minimization and pseudo-label learning techniques to address covariate shift. Experimental results demonstrate that OT3A significantly outperforms existing adaptation

Table 7: Performance of the OT3A approach and compared baselines on 6 datasets using TabTransformer. The best performance is in **Bold**. More experimental results are in the Supplemental Material.

| Method | HELOC | | | Voting | | | Diabetes | | |
|---|---|---|---|---|---|---|---|---|---|
| | Acc | bAcc | F1 | Acc | bAcc | F1 | Acc | bAcc | F1 |
| Source | 55.13 ±1.5 | 59.31 ±0.5 | 50.12 ±0.3 | 77.89 ±0.1 | 73.87 ±0.0 | 74.74 ±0.2 | **83.23** ±0.1 | 54.42 ±0.1 | 54.08 ±0.0 |
| PL | 54.26 ±0.1 | 58.72 ±0.3 | 51.44 ±0.3 | 77.25 ±0.4 | 72.97 ±0.3 | 73.53 ±0.5 | 83.00 ±0.3 | 52.39 ±0.1 | 50.32 ±0.2 |
| TTT | 54.15 ±0.5 | 58.21 ±0.4 | 49.31 ±0.6 | 77.18 ±0.5 | 74.10 ±0.5 | 73.64 ±0.2 | 83.10 ±0.0 | 52.13 ±0.2 | 52.41 ±0.1 |
| TENT | 54.35 ±2.3 | 57.24 ±5.8 | 49.93 ±2.6 | 78.07 ±0.6 | 74.05 ±0.6 | 74.16 ±0.3 | 83.12 ±0.0 | 52.00 ±0.0 | 52.20 ±0.0 |
| EATA | 55.07 ±0.1 | 57.25 ±0.2 | 50.02 ±0.6 | 78.03 ±0.1 | 76.20 ±0.5 | 74.79 ±0.1 | 83.00 ±0.3 | 52.12 ±0.4 | 52.05 ±0.1 |
| LAME | 49.15 ±0.6 | 56.05 ±0.0 | 50.10 ±0.6 | 73.50 ±0.1 | 74.60 ±0.4 | 74.80 ±0.1 | 82.54 ±0.1 | 52.83 ±0.2 | 52.56 ±0.6 |
| ODS | 54.10 ±0.6 | 57.00 ±0.0 | 51.14 ±0.2 | 77.50 ±2.1 | 74.60 ±0.5 | 74.80 ±0.0 | 82.26 ±0.1 | 54.16 ±0.1 | 52.06 ±0.1 |
| SAR | 54.32 ±2.6 | 56.74 ±0.9 | 48.18 ±2.9 | 77.13 ±0.6 | 74.20 ±0.5 | 73.79 ±0.1 | 82.98 ±0.2 | 53.48 ±0.4 | 52.81 ±0.9 |
| **OT3A** | **63.39** ±0.7 | **62.67** ±0.6 | **62.67** ±0.5 | **79.96** ±0.1 | **79.00** ±0.1 | **79.12** ±0.2 | 76.52 ±0.2 | **73.62** ±0.1 | **67.67** ±0.2 |

| Method | ASSISTments | | | Hospital Readmission | | | Childhood Lead | | |
|---|---|---|---|---|---|---|---|---|---|
| | Acc | bAcc | F1 | Acc | bAcc | F1 | Acc | bAcc | F1 |
| Source | 45.22 ±3.2 | 51.34 ±3.3 | 33.72 ±1.8 | 61.49 ±0.3 | 61.03 ±0.2 | 60.44 ±1.6 | 96.68 ±0.8 | 85.67 ±0.4 | 90.07 ±0.5 |
| PL | 44.12 ±0.4 | 50.41 ±0.3 | 31.32 ±0.5 | 60.29 ±0.2 | 60.00 ±0.0 | 57.83 ±0.6 | **96.93** ±0.4 | 86.85 ±0.1 | **91.07** ±0.2 |
| TTT | 44.16 ±1.3 | 50.02 ±0.6 | 35.92 ±0.2 | 60.46 ±0.3 | 59.46 ±0.0 | 59.62 ±0.1 | 96.83 ±0.2 | 85.00 ±0.0 | 90.90 ±0.0 |
| TENT | 40.87 ±0.3 | 46.41 ±0.3 | 33.99 ±0.2 | 61.12 ±0.3 | 60.15 ±0.3 | 53.75 ±1.0 | 91.93 ±0.0 | 84.48 ±0.0 | 90.15 ±0.0 |
| EATA | 45.86 ±0.2 | 50.81 ±0.2 | 36.42 ±0.1 | 61.36 ±0.3 | 60.16 ±0.3 | 53.68 ±0.9 | 91.93 ±0.0 | 84.98 ±0.0 | 90.02 ±0.0 |
| LAME | 45.12 ±0.2 | 48.79 ±0.1 | 31.40 ±0.1 | 60.05 ±0.1 | 58.90 ±0.3 | 46.69 ±1.4 | 91.93 ±0.0 | 50.00 ±0.0 | 47.90 ±0.0 |
| ODS | 45.12 ±0.2 | 51.30 ±0.1 | 32.08 ±0.1 | 61.39 ±0.1 | 60.12 ±0.3 | 59.25 ±1.4 | 91.93 ±0.0 | 50.00 ±0.0 | 47.90 ±0.0 |
| SAR | 45.86 ±0.2 | 50.81 ±0.1 | 36.42 ±0.1 | 60.38 ±0.2 | 57.89 ±0.3 | 58.48 ±0.9 | 97.08 ±0.0 | 86.58 ±0.0 | 91.00 ±0.0 |
| **OT3A** | **56.93** ±0.4 | **53.94** ±1.1 | **52.55** ±0.6 | **61.70** ±0.2 | **61.71** ±0.2 | **61.70** ±0.9 | 95.88 ±0.0 | **97.76** ±0.1 | 88.69 ±0.2 |

methods in terms of performance. However, our method also has some limitations. First, the application of OT3A is currently restricted to classification tasks, and extending it to other types of tasks (e.g., regression) remains challenging. Since OT3A depends on the generation of pseudo-labels, adapting it for continuous value predictions (such as regression tasks) is not straightforward. It is worth noting that similar challenges are faced by other methods as well, given the difficulty in effectively estimating prediction entropy for continuous outputs.

Secondly, OT3A cannot perform adaptation at the single-sample level. This is because some key steps in the method (such as determining pseudo-labels in the **C2P** stage through uncertainty estimation and performing label propagation) require batch-based computation, which makes the method inapplicable directly in single-sample inference scenarios. In the future, we will further explore instance-level adaptation methods and extend this research framework to a wider range of application scenarios.

## D.2 BROADER IMPACTS

Our research addresses the critical yet underexplored challenge of distribution shifts in tabular data, a problem that has not received sufficient attention. We believe that our approach can significantly enhance the performance of machine learning models in various industries by improving model adaptation to tabular data, thereby creating meaningful value in practical applications. Through our data-centric analysis in Section 2.2, we identify why existing TTA methods fail in the tabular domain and introduce a tabular-specific approach for handling label distribution shifts in Section 3.2. We hope this work will provide valuable insights for future research on test-time adaptation in tabular data. Additionally, by making our source code publicly available, we aim to support real-world applications across various fields, benefiting both academia and industry.

