# OpenReview forum: "Online Test-Time Adaptation in Tabular Data with Minimal High-Certainty Samples"
_ICLR.cc/2026/Conference — ICLR 2026 Conference Withdrawn Submission_

### Official Review · Reviewer_Fui1 · 2025-10-15

**Soundness:** 1
**Presentation:** 1
**Contribution:** 1
**Rating:** 2
**Confidence:** 5

**Summary:**

This paper proposed a Test-time adaptation method named OT3A. Based on the analysis of the imbalanced class, it leverages high-confidence and domain-consistent pseudo-labels to estimate and correct for target label distribution shifts. Additionally, it employs self-training and entropy minimization, guided by these confident samples, to adapt the model to the out-of-distribution test data. The experiment shows the effectiveness of OT3A

**Strengths:**

1. The analysis 2 is interesting and gives insights into tabular machine learning.
2. The experimental design is comprehensive and well-structured

**Weaknesses:**

1. This paper lacks novelty. Here is the detail:
- In the introduction section, line 42, this conclusion is proposed early in Observation 4 in [1].
- In the preliminary section, lines 85-88 represent the same as section 2.1 in [1]. Analysis 1, from lines 103 to 130, has already been reported in previous studies, and the analyses, experiments, and descriptions in this paper are consistent with prior research ( [1] Observation 1).
- In the method part,  the usage of the neighbor message and the correction of prediction in this study has been mentioned and applied in previous research [1] [2]. This paper only makes some minor adjustments in specific details, while the overall methodological framework remains similar to prior studies. **The description is also similar to earlier works  (line 186-188 is similar to section 3.1 in [1], line 195-201 is similar to section 3.3 in [2] ), indicating limited novelty.**
- **The paper shows a high degree of similarity to previously published work, with portions of the text being identical, which lacks references.** Here are the details besides the ones mentioned above:

a. Line 263-267 is similar to section 4.1-Evaluation Protocol in [1].

b. Line 274-280 is similar to section 4.1 in [2]

c. Line 698-715 **is the same as** E.3 in [2]

d. Line 726-731 **is the same as** E.1 MLP in [2]

---
2. Some parts of the paper are not clearly articulated. For example, in lines 60–64, the discussion shifts directly from *high/low-entropy samples* to *high-consistency samples* without explaining the connection between them. You only mention that *low-entropy samples* are beneficial for model convergence, but it remains unclear how *low-entropy samples* relate to *consistency samples*.
---
3. The symbol is not clear. It is hard to follow the methods with unclear mathematical symbols. For example, in section 3.2 line 224, what does S mean, and how does S come from? In addition, it is not clear whether kNN and RBF function jointly or independently in the proposed method.
---

In summary, this paper has a high similarity to previously published work, which lacks novelty, not meet the standard of ICLR.

---

[1] Zhou Z, Yu K Y, Guo L Z, et al. Fully test-time adaptation for tabular data. In: AAAI 2025

[2] Kim C, Kim T, Woo S, et al. Adaptable: Test-time adaptation for tabular data via shift-aware uncertainty calibrator and label distribution handler. In: NeurIPS Workshop on Table Representation Learning (NeurIPSW-TRL), 2024

**Questions:**

See Weakness above

---

### Official Review · Reviewer_9D9J · 2025-10-27

**Soundness:** 2
**Presentation:** 2
**Contribution:** 2
**Rating:** 2
**Confidence:** 4

**Summary:**

The paper studies online test-time adaptation (OTTA) for tabular classification and proposes OT3A, which (i) selects consistency-confidence points (C2P) via per-class confidence quantiles and local agreement, (ii) estimates/corrects label shift by combining batch predictions with label propagation over an affinity graph, and (iii) adapts to covariate shift using pseudo-label self-training plus entropy minimization; updated per incoming batch. Experiments on six Tableshift datasets with MLP/TabTransformer backbones show sizable gains in balanced accuracy and macro-F1 over several TTA baselines.

**Strengths:**

1. Consistent empirical gains across multiple datasets/architectures.

**Weaknesses:**

1. Missing tabular-TTA baselines and prior art that already tackle label+covariate shift. The paper compares against generic TTA methods (PL, TTT, TENT, EATA, SAR, LAME, ODS) but omits tabular-specific TTA that are close in spirit: AdaptTable [1], FTAT [2].

1. Novelty claims around “co-existence of label & covariate shift + class imbalance” are not new.
Both AdapTable and FTAT emphasize exactly these phenomena for tabular TTA and build methods to handle them; the current paper should reposition its contribution as a specific C2P-driven estimation/propagation strategy rather than the first to recognize the setting.


Minor weaknesses

1. The label shift correction method requires source label distribution.

1. Figure axis legends have small font sizes.


[1] Kim, Changhun, et al. "Adaptable: Test-time adaptation for tabular data via shift-aware uncertainty calibrator and label distribution handler." arXiv preprint arXiv:2407.10784 (2024).

[2] Zhou, Zhi, et al. "Fully test-time adaptation for tabular data." Proceedings of the AAAI Conference on Artificial Intelligence. Vol. 39. No. 21. 2025.

**Questions:**

1. What is the rationale of the loss function design, consisting both CE and Entropy loss?

1. Would this method applicable to vision dataset with label+covariate shift?

Minor

1. Line 180: Is this a typo? "where τc is the value corresponding to the *τc quantile* in the consistency distribution"

---

### Official Review · Reviewer_zHkM · 2025-10-27

**Soundness:** 1
**Presentation:** 1
**Contribution:** 1
**Rating:** 2
**Confidence:** 5

**Summary:**

* Tabular datasets often exhibit distribution shifts (label/covariate shift, class imbalance) when deployed in real-world environments, causing significant drops in model performance. Existing TTA methods from vision and NLP domains are suboptimal when applied to tabular data.
* Proposes OT3A method - which selects a subset of locally consistent and highly confident subset of test data. Then, it is calibrated, by aligning the estimated label distributions between source and target.
* Evaluated upon multiple dataset, showing performance gains upon prior methodologies.

**Strengths:**

.

**Weaknesses:**

Questions regarding core contribution
* Most of the statements made in the paper align with AdapTable(https://arxiv.org/abs/2407.10784). I cannot see the difference in its statements made in 2.2 Problem Analysis, showing the exact same findings once more.
Questions regarding novelty of proposed method.
* This also tracks shifted label distributions using high-confidence predictions, and uses acovariance matrix to correct the bias in label distribution estimation. Why does it not compare it with this methodology? (https://arxiv.org/html/2412.10871v1). Utilization of uncertainty is a well-founded method(mixture of it as well). Are the authors claim solely rise from the application of these methods to the tabular domain?
Questions regarding source model used.
* The AdamW optimizer is used with a learning rate of 0.01 and a weight decay of 0.01. -> this seems to be an EXTREME fixation upon the source model. I wish the authors could train a variety of source models, select one based upon the best validation performance and use it for test-time adaptation.
* I believe most of the performance gains are due to the mitigation of different label distributions. for a good comparison, the authors should train their source models upon a "balanced" dataset, and log their performance gains. For datasets where minorities are under-represented, there are a multiple serires of upsampling techniques to ensure balanced training. (https://arxiv.org/abs/2012.01696)

**Questions:**

refer to above.

---

### Official Review · Reviewer_wrEQ · 2025-10-30

**Soundness:** 2
**Presentation:** 3
**Contribution:** 2
**Rating:** 2
**Confidence:** 5

**Summary:**

- This paper introduces OT3A (Online Test-Time Tabular Adaptation), a framework designed for test-time adaptation in tabular domains—a setting largely overlooked.

- OT3A dynamically adapts models during inference using high-confidence pseudo-labels to correct label shift and entropy minimization to handle covariate shift.

- The method targets real-world tabular challenges such as class imbalance and multi-type shifts, showing consistent improvements across benchmarks like HELOC, Voting, and Diabetes.

**Strengths:**

- This paper tackles online test-time adaptation specifically for tabular data, an underexplored and practically important problem in fields like healthcare and finance.

- Combines pseudo-labeling for label shift and entropy minimization for covariate shift in a unified and interpretable way.

- This paper also addresses real-world issues such as class imbalance and complex mixed-type shifts, improving model reliability in deployment scenarios.

**Weaknesses:**

### **Motivation**
- The idea of test-time adaptation for tabular data is not new. Prior works such as TabLog [1], FTAT [2], AdapTable [3], and [4] have already investigated similar directions. The current paper fails to cite AdapTable despite its clear conceptual overlap, raising concerns about potential plagiarism.

- Moreover, the proposed components—uncertainty estimation, label distribution correction, and entropy minimization—have all been explored in existing tabular TTA frameworks (e.g., AdapTable, FTAT, AdaTab). In particular, using entropy minimization for covariate shift and label-shift calibration for distribution correction is standard practice in many previous works, and thus provides no technical novelty here.

- The paper also claims novelty in handling the co-existence of covariate and label shifts, but this is a well-known issue already discussed multiple times in the vision TTA literature. The contribution in this respect is incremental at best.

---

### **Methodology**

- The proposed method overlaps heavily with AdapTable. The use of the maximum difference between the top two logits as an uncertainty measure is identical, and the label distribution correction in line 198 replicates the same mechanism. These similarities appear to go beyond inspiration and verge on textual or conceptual plagiarism.

- Additionally, the use of Euclidean distance to measure differences between high-dimensional tabular features is inappropriate. Tabular data often mix heterogeneous feature types and scales, making Euclidean metrics mathematically and practically meaningless without normalization or feature weighting.

---

### **Experiments**

- All experiments are performed only on binary classification datasets from TableShift. This setup is insufficient to validate the method’s effectiveness. Binary tasks make label-shift correction artificially easy, since biasing predictions toward the majority class can inflate accuracy. Evaluation on multi-class datasets is essential to demonstrate robustness.

- The reported batch sizes (250–2000) are unrealistic for tabular adaptation experiments. Standard batch sizes rarely exceed 128 or 256. Using such large batches distorts learning dynamics and makes results incomparable to prior baselines.

- The component-wise ablation is also weakly analyzed. Several ablations even outperform the full method, suggesting the model’s design lacks internal consistency.

---

### **Writing**

-  The notation of the Affinity Matrix (\hat{s}_{ij}) contains a typo.
- The ablation table misuses bold formatting, highlighting configurations that are not the best-performing.

---

### **References**
- [1] Ren et al. "Test-Time Adaptation for Tabular Data Using Logic Rules." ICML, 2025.
- [2] Zhou et al. "Fully Test-time Adaptation for Tabular Data." AAAI, 2025.
- [3] Kim et al. "AdapTable: Test-Time Adaptation for Tabular Data via Shift-Aware Uncertainty Calibrator and Label Distribution Handler." NeurIPS Workshop on Table Representation Learning, 2024.
- [4] Zeng et al. "LLM Embeddings Improve Test-time Adaptation to Tabular Y|X-Shifts." NeurIPS Workshop on Table Representation Learning, 2025.

**Questions:**

* How exactly was Figure 1 computed? The paper provides no details on the data or metric used to construct this figure.

---

### Note · Authors · 2025-11-12

I have read and agree with the venue's withdrawal policy on behalf of myself and my co-authors.